# OpenReview forum: "Exposing Hidden Biases in Text-to-Image Models via Automated Prompt Search"
_ICLR.cc/2026/Conference — Submitted to ICLR 2026_

### Official Review · Reviewer_jBhd · 2025-10-15

**Soundness:** 2
**Presentation:** 2
**Contribution:** 2
**Rating:** 2
**Confidence:** 4

**Summary:**

This paper investigates the issue of social biases in TTI models. It point out that existing bias detection methods are either limited by the scope of manual test sets or generate uninterpretable adversarial text. To address this, the paper proposes an automated framework named Bias-Guided Prompt Search (BGPS). This framework combines an LLM with attribute classifiers that operate on the internal activations of the TTI model. The core mechanism involves using the bias signals from the classifiers to steer the LLM's decoding process, thereby automatically discovering prompts that maximize bias exposure while maintaining natural language fluency. The authors conducted experiments on Stable Diffusion 1.5 and a debiased model. The results show that BGPS can find hidden, undocumented biases and reveal that significant biases persist even in models that have undergone debiasing efforts.

**Strengths:**

**1. Well-motivated problem and a practical objective**

 The objective of developing an automated framework to audit TTI models for fairness and safety is of high practical importance for the responsible deployment of generative AI.

**2. Generation of interpretable, human-readable prompts**

A primary strength of the proposed method is its ability to generate human-readable and interpretable prompts. The paper correctly identifies the limitations of gradient-based optimization methods, which often produce nonsensical and un-interpretable text. By leveraging a large language model, the BGPS framework successfully generates fluent, natural language prompts.

**Weaknesses:**

**1. Insufficient Experimental Scope and Lack of Generalizability**

The paper's claims of providing a generalizable framework are unsubstantiated due to a severely limited experimental scope. The evaluation is confined entirely to a single, model, Stable Diffusion 1.5. The failure to test the method on any other diverse, modern (SDXL, Flux), or transformer-based (DALL-E 3, SD3) models means it is impossible to assess whether the framework is truly general-purpose.

**2. Use of Outdated and Unspecified Models**

The core components used in the experiments are outdated and poorly specified, making the work feel disconnected from the current models in 2025. Beyond the use of the old SD 1.5 model, the paper relies on an aging LLM (Mistral-7B) and, critically, fails to specify which version was used. This omission is a significant flaw that severely harms the reproducibility of the research.

**3. Minimal Method Novelty**

The paper's methodological contribution is largely incremental. The proposed framework is a straightforward application of existing principles in guided decoding, with basic prompt engineering for LLMs. As such, the work's novelty from a methodological standpoint is limited.

**4. No New Scientific Insight**

While the method is effective at finding biases, the experiments primarily succeed in rediscovering well-documented and highly intuitive social biases and stereotypes that have been long established in both sociology and prior AI fairness literature. The paper offers little to no new scientific insight into the nature of bias, serving more as a confirmation of known issues.

**5. Poor Presentation Quality of Figures and Captions**

Figures 2 and 3, which are crucial for illustrating the method's outputs, are hindered by poorly formatted captions. The prompt texts are broken across lines in unnatural and confusing ways, which severely impairs readability and forces the reader to decipher the intended meaning. This detracts from the paper's professionalism and weakens the impact of its results.

**Questions:**

**1.** Given that the current experiments are limited to outdated models, can the authors provide any new results on contemporary SOTA models to substantiate the framework's claimed generalizability?

**2.** Beyond confirming well-documented social biases, did the authors' method uncover any counter-intuitive or genuinely novel insights into the nature of bias in TTI models?

**3.** I strongly encourage the authors to commit to reformatting Figures 2 and 3, as their current layout severely hinders the readability of the paper's core qualitative evidence.

---

> ### Author Response · Authors · 2025-11-25
>
> **1. Insufficient Experimental Scope and Lack of Generalizability**
>
> To assess how our method works in newer diffusion models, we conducted experiments on SD2.1, SDXL and DeepFloyd IF. We found that BGPS could reliably find prompts that amplify biases in all three models. As stated in our response to reviewer **ZJ5G**: While different diffusion models exhibited varying levels of baseline gender bias, with the manual curated test set having $\sim50 \%$ male in SD1.5, $\sim60 \%$ in SDXL, up to $\sim68 \%$ in DeepFloyd IF, BGPS succeeded in increasing male/female proportion in all models, with minimal increases in Perplexity.
>
> All new diffusion model experiment results are in Table 7 in the Appendix. As our method currently relies on the h-space [1] it is applicable to unet-based diffusion architectures. Adapting our method to transformer architectures would be non-trivial, so we leave it as future work.
>
> **2. Use of Outdated and Unspecified Models**
>
> We thank the reviewer for pointing this out. We have added the exact version of the LLM used for the experiments to the paper text (Mistral-7B-0.2). In order to ensure that our work is reproducible, we provide a detailed description of all external models used for our experiments in the Appendix, along with links to the respective checkpoints. We also will be releasing the code for all experiments upon acceptance of our work.
>
> Our method relies on the LLM to effectively determine the language prior for our objective, which is natural-sounding and gender neutral text descriptions of images. To ablate the effect of the LLM prior, we have added two new LLM experiments, with Qwen-3-8B, a similarly sized model, that was trained with a large amount of multilingual data, and LLaMA-3.2-1B, a smaller model. Both LLMs have been released in 2025 (April and September respectively). Results of the new LLM experiments are in Tables 1 and 2 in the main paper.
>
> Regarding the diffusion models used, all the models in our paper are widely used in recent publications for top-tier machine learning conferences [2,3,4,5].
>
> ---
>
> [1] Kwon, Mingi, Jaeseok Jeong, and Youngjung Uh. "Diffusion Models Already Have A Semantic Latent Space." In The Eleventh International Conference on Learning Representations.
>
> [2] Giannis Daras, Weili Nie, Karsten Kreis, Alex Dimakis, Morteza Mardani, Nikola Kovachki, and
> Arash Vahdat. Warped diffusion: Solving video inverse problems with image diffusion models.
> Advances in Neural Information Processing Systems, 37:101116–101143, 2024
>
> [3] Viacheslav Surkov, Chris Wendler, Mikhail Terekhov, Justin Deschenaux, Robert West, and Caglar
> Gulcehre. Unpacking sdxl turbo: Interpreting text-to-image models with sparse autoencoders. In
> Mechanistic Interpretability for Vision at CVPR 2025 (Non-proceedings Track), 2025.
>
> [4] Wei Xu, Kangjie Chen, Jiawei Qiu, Yuyang Zhang, Run Wang, Jin Mao, Tianwei Zhang, and Lina
> Wang. Automated red teaming for text-to-image models through feedback-guided prompt itera-
> tion with vision-language models. In Proceedings of the IEEE/CVF International Conference on
> Computer Vision, pp. 18575–18584, 2025.
>
> [5] Nick Stracke, Stefan Andreas Baumann, Kolja Bauer, Frank Fundel, and Bj¨orn Ommer. Cleandift:
> Diffusion features without noise. In Proceedings of the IEEE/CVF Conference on Computer
> Vision and Pattern Recognition (CVPR), pp. 117–127, June 2025.

---

> > ### Author Response · Authors · 2025-11-25
> >
> > **3. Minimal Method Novelty**
> >
> > Please refer to our answer to reviewer **Sk3D** regarding our method's technical novelty:
> >
> > Our work is, to the best of our knowledge, the first framework that explicitly formulates the problem of automatic discovery of biased prompts for text-to-image (TTI) diffusion models, under the dual constraints that prompts (i) remain human-like and attribute-neutral and (ii) are evaluated through the \emph{image-space} behaviour of a downstream TTI model. While the idea of using an LLM with an external score has appeared before (e.g., for sentiment control, safety, or TTI prompt inversion), existing methods do not integrate diffusion-model attribute classifiers into a joint probabilistic objective as we do.
> >
> > Regarding the phrasing "no strong technical insight or theoretical analysis”, our contribution is primarily methodological and empirical: we provide (i) a clear probabilistic formulation of bias-guided prompt search, (ii) an efficient algorithm that can be applied across multiple diffusion models and debiasing methods,(iii) extensive experiments on four different widely used diffusion models and two recent debiasing methods that demonstrate the effectiveness of BGPS in discovering novel biases in TTI models, and (iv) an in-depth analysis of the resulting prompts and bias mechanisms (new Section 4.4 Analysis of Biased Prompts). We believe that this combination of a principled objective, a practically effective search procedure, extended experiments and systematic analysis constitutes meaningful  novelty, even if it does not come with formal convergence or generalization guarantees.
> >
> > **4. No New Scientific Insight**
> >
> > Indeed, most biases discovered by BGPS are already well established by both sociology and AI fairness literature, which is what **motivated** us to develop a method for discovering them. Furthermore, the fact that large drivers of biased behaviour in image generation models reflects well known social biases is expected, as text-to-image models inherit and amplify these biases largely through their training data [1].
> >
> > Although not intended as a bias evaluation framework, BGPS fills an important gap in the field of automatic bias assessment.
> > While most works in the literature identify occupational bias in TTI models by simple prompts like "a doctor" or "a photo of a doctor" [2,3], BGPS enables the generation of categories *in context*, finding that "doctor with compassionate eyes" is female-biased, while "doctor with serious expression" is male-biased. Recent works like [4] that use more complex prompts use extensive crowdsourcing to gather candidate prompts, which can be very labor-intensive.
> >
> > Furthermore, to systematically find bias mechanisms and enhance our paper's scientific insights into the nature of bias in text-to-image models, we have added a new section, *“Analysis of Biased Prompts”* (Sec. 4.4), where we systematically analyze the prompts discovered by BGPS and compare them to the LLM-only baseline. This section reveals a number of insights into how bias is propagated by BGPS.
> >
> > First, **bias arises mainly through *new* lexical cues rather than amplifying existing ones.** We find that BGPS does not primarily work by increasing the frequency of already male-coded words compared to the llm baseline. Instead, it increases bias mainly by *injecting* previously unused words that have strong gender associations in the TTI model (see Figure 4).
> >
> > Second, by visualizing the words more strongly associated with biased prompts (see Figure 4), we can get a good understanding of how bias is propagated in text-to-image models.
> >
> > ---
> >
> > [1] Preethi Seshadri, Sameer Singh, and Yanai Elazar. The bias amplification paradox in text-to-image
> > generation. In Proceedings of the 2024 Conference of the North American Chapter of the Associ-
> > ation for Computational Linguistics: Human Language Technologies (Volume 1: Long Papers),
> > pp. 6367–6384, 2024.
> >
> > [2] Ranjita Naik and Besmira Nushi. Social biases through the text-to-image generation lens. In Pro-
> > ceedings of the 2023 AAAI/ACM Conference on AI, Ethics, and Society, pp. 786–808, 2023.
> >
> > [3] Sasha Luccioni, Christopher Akiki, Margaret Mitchell, and Yacine Jernite. Stable bias: Evaluating
> > societal representations in diffusion models. Advances in Neural Information Processing Systems,
> > 36:56338–56351, 2023.
> >
> > [4] Leander Girrbach, Stephan Alaniz, Genevieve Smith, and Zeynep Akata. A large scale analysis of
> > gender biases in text-to-image generative models. arXiv preprint arXiv:2503.23398, 2025.

---

### Official Review · Reviewer_zJ5g · 2025-10-21

**Soundness:** 2
**Presentation:** 3
**Contribution:** 3
**Rating:** 6
**Confidence:** 5

**Summary:**

This paper automatically discovers interpretable prompts that maximize bias exposure while remaining natural. It optimizes a joint objective involving LLMS for naturalness and an attribute classifier on the middle block representations of the Stable Diffusion Model. They utilise a beam search algorithm to search through the prompts that maximize the joint objective. All the experiments are performed on SD1.5. Auditing is also performed for one of the existing debiasing methods.

**Strengths:**

This paper tackles the problem of exposing biases in T2I models using realistic, neutral-sounding prompts, which is an important problem that helps in auditing large-scale T2I models.

**Weaknesses:**

1. Results are reported primarily on SD-1.5. Since the method depends on UNet architectures, the evaluation should be extended to SD-2.1 and SDXL to assess robustness on newer, stronger models.
2. Given the paper’s focus on residual bias in “debiased” models, the audit should also include recent debiasing methods, e.g., ITIGen, which operates in the prompt space, to strengthen the generality of the claims.
3. The paper uses simple, clean, single-person prompts. In order to test the applicability to real-world settings, it is important to evaluate real-world prompts such as multi-person scenes (define how gender is assigned) or long/ambiguous prompts.
 4. Because an LLM is used as the prompt prior, its own biases can seep into the pipeline. How sensitive is the approach to LLMs used? It is important to include sensitivity to different LLM priors and an ablated/weaker prior to measure this effect.

**Questions:**

1. Can you provide more qualitative results and corresponding prompts for different races?
2. What happens when the attributes are very complex in this setup? How much does the performance of the attribute classifier affect the whole pipeline?

---

> ### Author Response · Authors · 2025-11-25
>
> **1. Results are reported primarily on SD-1.5. Since the method depends on UNet architectures, the evaluation should be extended to SD-2.1 and SDXL to assess robustness on newer, stronger models.**
>
> We have added additional experiments for three more Unet-based diffusion models for the gender attribute: latent diffusion models SD2.1, SDXL, and DeepFloyd IF, a diffusion model that works directly in the pixel space. For DeepFloyd IF we implemented BGPS for the stage I model, and evaluated on images upscaled with the full cascaded architecture. Results of the experiments are in Table 7 in the Appendix.
>
> While different diffusion models exhibited varying levels of baseline gender bias, with the manual curated test set having $\sim50 \%$ male in SD1.5, $\sim60 \%$ in SDXL, up to $\sim68 \%$ in DeepFloyd IF, BGPS succeeded in increasing male/female proportion in all models, with minimal increases in Perplexity.
>
> **2. Given the paper’s focus on residual bias in “debiased” models, the audit should also include recent debiasing methods, e.g., ITIGen, which operates in the prompt space, to strengthen the generality of the claims.**
>
> We have identified three main categories of debiasing methods in the literature, which are *fineteuning* methods, *activation steering* methods, and *prompt manipulation* methods.
> In our initial submission we audited the method proposed by [1], that finetunes the diffusion model's text encoder. In our rebuttal, in order to strengthen the generality of our method, we have included experiments on DiffLens [2], a recent framework that uses sparse autoencoder features to steer the diffusion unet towards fair image generation (results are in Tables 1 and 2 in the main paper). We found that BGPS consistently succeeds in discovering biased prompts for both finetuning and activation steering methods.
>
> Given that our method's goal is to generate prompts, we have opted to not include prompt-space debiasing methods in our auditing scope. This choice is due to the fact that most prompt manipulation methods attach explicit mention to the protected attribute in the prompt, in order to enforce the target distribution. For example, [3] debiases prompts by adding modifiers such as "white/black" or "male/female" after a profession to control race and gender attributes, while [4] guides generation with the embeddings for "male person" and "female person". ITI-GEN uses learned prompt embeddings for each attribute, mapped to special tokens (in the style of textual inversion), in order to steer the prompt towards the desired attribute. In contrast, we require all prompts generated by BGPS to be attribute-neutral, in order to discover novel or overlooked biases.
>
> ---
>
> [1] Xudong Shen, Chao Du, Tianyu Pang, Min Lin, Yongkang Wong, and Mo-
> han Kankanhalli. Finetuning text-to-image diffusion models for fairness. In
> The Twelfth International Conference on Learning Representations, 2024.
>
> [2] Yingdong Shi, Changming Li, Yifan Wang, Yongxiang Zhao, Anqi Pang, Sibei Yang, Jingyi Yu, and
> Kan Ren. Dissecting and mitigating diffusion bias via mechanistic interpretability. In Proceedings
> of the Computer Vision and Pattern Recognition Conference, pp. 8192–8202, 2025.
>
> [3] Colton Clemmer, Junhua Ding, and Yunhe Feng. Precisedebias: An automatic prompt engineering
> approach for generative ai to mitigate image demographic biases. In Proceedings of the IEEE/CVF
> Winter Conference on Applications of Computer Vision (WACV), pp. 8596–8605, January 2024.
>
> [4] Felix Friedrich, Manuel Brack, Lukas Struppek, Dominik Hintersdorf, Patrick Schramowski, Sasha
> Luccioni, and Kristian Kersting. Fair diffusion: Instructing text-to-image generation models on
> fairness. arXiv preprint at arXiv:2302.10893, 2023.

---

> ### Author Response · Authors · 2025-11-25
>
> **3. The paper uses simple, clean, single-person prompts. In order to test the applicability to real-world settings, it is important to evaluate real-world prompts such as multi-person scenes (define how gender is assigned) or long/ambiguous prompts.**
>
> We thank the reviewer for the insightful suggestion. We stress tested our method by incorporating both scenarios at the same time, asking the LLM to generate descriptions of multiple persons, while increasing the maximum token limit for the generation. This resulted in prompts almost twice as long as the original experiment prompts: 22.5 ± 2.5 words compared to 13 ± 1.5 for original occupation experiment. The evaluation pipeline detects all faces in generated images and produces gender classification logits, while the total male/female proportion is the average proportion across all generated faces. Results for the experiment can be seen in Table 9 in the Appendix, and sample prompts and images in Figure 7.
>
> BGPS is successful in generating both male- and female- biased prompts in this scenario without degrading prompt quality. We report proportions for BGPS with $\lambda=10,100$ in the Appendix, as well as generated prompts and images for this scenario. We also report the average number of detected faces which is greater than 3 for all experiments.
>
> **4. Because an LLM is used as the prompt prior, its own biases can seep into the pipeline. How sensitive is the approach to LLMs used? It is important to include sensitivity to different LLM priors and an ablated/weaker prior to measure this effect.**
>
> As discussed in the Limitations section of our paper, we acknowledge that the language model prior can influence the biases that are present in the generated prompts. To investigate the effects of the LLM prior, we conducted two additional experiments, both using recent (this year's) LLMs, one with a comparable parameter size to our baseline Mistral 7B v0.2 model, Qwen 3 8B, and another much smaller model, LLaMA 3.2 1B. Results are found in Tables 1,2 in the main paper.
> We added the following comments on the experiments in the paper:
>
>  We observe that our method can reliably bias prompts generated by all language models. It is interesting that the smaller LLaMA model, while comparable to larger models in perplexity, has a higher percentage of explicitly gendered prompts, bypassing our instructions for gender-neutral generation. Furthermore, upon inspection we can see that the quality of many prompts is compromised, even without using BGPS, with about one third of the prompts explicitly mentioning text-to-image generation models, while being instructed to only generate the prompt. This is indicative of the known limitation of smaller models to follow instructions [1,2].
>
> **Q1. Can you provide more qualitative results and corresponding prompts for different races?**
>
> We have included more qualitative results for prompts biased towards black people in Figure 6 in the Appendix. As all models are predominantly white-biased, we do not show explicitly white-biased images. Prompts and images with black bias perpetuate many known biases against black persons, depicting jazz musicians, "urban" artists, criminals as well as law enforcement.
>
> **Q2. What happens when the attributes are very complex in this setup? How much does the performance of the attribute classifier affect the whole pipeline?**
>
> The attribute classifiers we use are simple linear heads trained on the diffusion unet's middle layer activations. Gender and race attribute classifiers used in the experiments were taken from [3], where they report their accuracy to be over 90\% for activations from all diffusion timesteps. It is expected that classifiers with low accuracy would deteriorate our method's performance.
> It would be helpful if the reviewer could clarify what they mean by the phrase "very complex attributes", to specify our response further.
>
> ---
>
> [1] Zhihan Zhang, Shiyang Li, Zixuan Zhang, Xin Liu, Haoming Jiang, Xianfeng Tang, Yifan Gao,
> Zheng Li, Haodong Wang, Zhaoxuan Tan, et al. Iheval: Evaluating language models on follow-
> ing the instruction hierarchy. In Proceedings of the 2025 Conference of the Nations of the Amer-
> icas Chapter of the Association for Computational Linguistics: Human Language Technologies
> (Volume 1: Long Papers), pp. 8374–8398, 2025.
>
> [2] Rudra Murthy, Praveen Venkateswaran, Prince Kumar, and Danish Contractor. Kcif: Knowledge-
> conditioned instruction following, 2025. URL https://arxiv.org/abs/2410.12972.
>
> [3]  Yingdong Shi, Changming Li, Yifan Wang, Yongxiang Zhao, Anqi Pang, Sibei Yang, Jingyi Yu, and
> Kan Ren. Dissecting and mitigating diffusion bias via mechanistic interpretability. In Proceedings
> of the Computer Vision and Pattern Recognition Conference, pp. 8192–8202, 2025.

---

### Official Review · Reviewer_Sk3D · 2025-10-27

**Soundness:** 2
**Presentation:** 2
**Contribution:** 2
**Rating:** 4
**Confidence:** 4

**Summary:**

This paper introduces Bias-Guided Prompt Search (BGPS), an automated method that uses LLM and attribute classifiers to discover text prompts for exposing hidden social biases (e.g., gender, race) in text-to-image models. The proposed method effectively uncovers subtle, context-driven biases that standard evaluations miss and produces more natural prompts than gradient-based alternatives.

**Strengths:**

1. The proposed method can discover subtle and previously undocumented biases, which expands the evaluation space beyond curated datasets.

2. Compared with gradient-based methods, the proposed method can generate more natural text.

**Weaknesses:**

1. The scope of biased attributes evaluated (gender, race) is limited by the classifiers used, though the method is generalizable.

2. Although the generated prompts look more natural than the prompts generated by gradient-based methods, as shown in figure 1, the generated prompts are still not very natural (not like common prompts written by human)

3. The technical novelty is a little limited. There is neither strong technical insight or theoretical analysis.

**Questions:**

Please refer to the weakness part.

---

> ### Author Response · Authors · 2025-11-25
>
> **1. The scope of biased attributes evaluated (gender, race) is limited by the classifiers used, though the method is generalizable.**
>
> We acknowledge that the implementation of our method relies on the use of external attribute classifiers. However, this does not limit the applicability of our method to a variety of attributes. The attribute classifiers used in our paper consist of lightweight linear heads that can be trained quickly and efficiently. As they are trained on model activations, they do not require an external dataset of images for training, but only a small set of 24 prompts, created by permuting attribute labels ("A photo of a young/old white/black/indian/asian male/female person").
>
> Furthermore, it is straightforward to train classifiers for different attributes using the recipe found in [1][2]. We will include scripts for training attribute classifiers for the additional models used in our experiments (Stable Diffusion 2.1, SDXL, DeepFloyd IF) and arbitrary attributes when we publicly release the paper's code.
>
> To show how our method can be easily expanded to different attributes, after training a new binary age classifier, we have included an experiment for age biasing (see Appendix Section F.3). BGPS succeeded in biasing prompts towards adult and old persons.
>
> **2. Although the generated prompts look more natural than the prompts generated by gradient-based methods, as shown in figure 1, the generated prompts are still not very natural (not like common prompts written by human)**
>
> To measure prompt naturalness, we utilize the Perplexity metric, which has been widely used to score text naturalness and fluency in the literature [3,4,5]. While it is true that there is a trade-off between prompt perplexity and biasing strength, BGPS can increase prompt bias with minimal or no perplexity increase from the baseline LLM generation most of the time.
>
> The other reviewers have also emphasized as key strengths of our method, how biased prompts generated by BGPS are "realistic", "neutral-sounding" (**zJ5g**), "human-readable and interpretable", and also how "the BGPS framework successfully generates fluent, natural language prompts" (**jBhd**).
>
> ---
>
> [1] Yingdong Shi, Changming Li, Yifan Wang, Yongxiang Zhao, Anqi Pang, Sibei Yang, Jingyi Yu, and
> Kan Ren. Dissecting and mitigating diffusion bias via mechanistic interpretability. In Proceedings
> of the Computer Vision and Pattern Recognition Conference, pp. 8192–8202, 2025.
>
> [2] Rishubh Parihar, Abhijnya Bhat, Abhipsa Basu, Saswat Mallick, Jogendra Nath Kundu, and
> R Venkatesh Babu. Balancing act: distribution-guided debiasing in diffusion models. In Proceed-
> ings of the IEEE/CVF conference on computer vision and pattern recognition, pp. 6668–6678,
> 2024
>
> [3] Donghoon Kim, Minji Bae, Kyuhong Shim, and Byonghyo Shim. Visually guided decoding:
> Gradient-free hard prompt inversion with language models. In International Conference on
> Learning Representations (ICLR), 2025
>
> [4] Subham Sahoo, Marianne Arriola, Yair Schiff, Aaron Gokaslan, Edgar Marroquin, Justin Chiu,
> Alexander Rush, and Volodymyr Kuleshov. Simple and effective masked diffusion language
> models. Advances in Neural Information Processing Systems, 37:130136–130184, 2024.
>
> [5] Xiao Zhang and Ji Wu. Dissecting learning and forgetting in language model finetuning. In The
> Twelfth International Conference on Learning Representations, 2024

---

> > ### Author Response · Authors · 2025-11-25
> >
> > **3. The technical novelty is a little limited. There is neither strong technical insight or theoretical analysis.**
> >
> > Our work is, to the best of our knowledge, the first framework that explicitly formulates the problem of automatic discovery of biased prompts for text-to-image (TTI) diffusion models, under the dual constraints that prompts (i) remain human-like and attribute-neutral and (ii) are evaluated through the *image-space* behaviour of a downstream TTI model. While the idea of using an LLM with an external score has appeared before (e.g., for sentiment control, safety, or TTI prompt inversion), existing methods do not integrate diffusion-model attribute classifiers into a joint probabilistic objective as we do.
> >
> > Regarding the phrasing "no strong technical insight or theoretical analysis”, our contribution is primarily methodological and empirical: we provide (i) a clear probabilistic formulation of bias-guided prompt search, (ii) an efficient algorithm that can be applied across multiple diffusion models and debiasing methods,(iii) extensive experiments on four different widely used diffusion models and two recent debiasing methods that demonstrate the effectiveness of BGPS in discovering novel biases in TTI models, and (iv) an in-depth analysis of the resulting prompts and bias mechanisms (new Section 4.4 Analysis of Biased Prompts). We believe that this combination of a principled objective, a practically effective search procedure, extended experiments and systematic analysis constitutes meaningful  novelty, even if it does not come with formal convergence or generalization guarantees.

---

### Author Response · Authors · 2025-11-25

We would like to thank the reviewers for the insightful comments, that helped us significantly improve our paper during the rebuttal process.

The three reviewers pointed out the following **strengths** of our paper:

* Reviewers **zJ5g** and **jBhd** highlighted the *significance* of our work ("an important problem", "Well-motivated problem...of high practical importance for the responsible deployment of generative AI")
* Reviewers **Sk3D** and **jBhd** recognized the *effectiveness* of our method to produce biased prompts in text-to-image models (**Sk3D**: "The proposed method effectively uncovers subtle, context-driven biases that standard evaluations miss...and previously undocumented biases, which expands the evaluation space beyond curated datasets."  **jBhd**: "The results show that BGPS can find hidden, undocumented biases...the method is effective at finding biases...")
* Reviewers **zJ5g** and **jBhd** agreed that our method *succeeds in producing natural, interpretable prompts* ("using realistic, neutral-sounding prompts", "A primary strength of the proposed method is its ability to generate human-readable and interpretable prompts.", "the BGPS framework successfully generates fluent, natural language prompts"), with reviewer **Sk3D** agreeing that "Compared with gradient-based methods, the proposed method can generate more natural text.".

## Rebuttal Summary

### Significantly expanded experimental section

Our main contribution in this rebuttal is the *extensive experiments* we have conducted to prove our method's effectiveness and capability of generalization.


* We audit DiffLens, a second recent debiasing method, discovering biases in both model finetuning debiasing and activation steering methods.
* We compare debias method auditing for two additional LLM language priors, Qwen3 8B and Llama 3.2 1B.
* We extend our method to three more unet-based diffusion models, SD2.1, SDXL and DeepFloyd IF showing that it is effective in discovering biases in all of them.
* We stress test our method with a new scenario using multiple persons and longer prompts, indicating its effectiveness on real-world scenarios.
* We train an age attribute classifier and successfully discover biases for adult/old persons.
* We conduct additional ablations on method hyperparameters $K$ and $T'$.


### Added bias mechanism analysis

We have strengthened our initial claims of discovering novel biases by conducting a systematic analysis of prompts generated by BGPS to (i) clarify our method's mechanism of action for producing biased prompts and (ii) provide insight into the nature of biases discovered by BGPS. To this extent we have found that:

* BGPS consistently introduces novel terms associated with the bias attribute, that were not present in the LLM baseline.
* By visualizing the terms associated with high levels of bias, we gain insight into how bias manifests in the text-to-image model.


### Added extensive qualitative results

We have included a large number of generated prompts for different biasing attributes, along with images showing model generation.

### Improved figure and text readability

We have significantly improved the readability and fluency of most of the paper's figures and tables, to ensure that results are easier to interpret and visually consistent throughout the manuscript.

---

### Author Response · Authors · 2025-12-02
**Comment to New AC**

ICLR distinguishes itself among other major conferences by its extended discussion period, which enables fruitful interactions between authors and reviewers and results in greatly improved works.

*Given the premature ending of the discussion period, we would like to emphasize that although we have used the rebuttal period to significantly supplement the initial submission (as per ICLR's philosophy & taking into account all of the points made by the reviewers), we had not received any reviewer comments to our rebuttal at the time of freezing the discussion.*

In light of the above, we are writing this comment to describe how we addressed the reviewer's concerns with specific additions to the final manuscript.

In the rebuttal revision, we addressed all the requests and added a significant amount of new material.

Specifically, for Reviewer **zJ5g** we address **all of their requests**:

* To extend our method to newer and stronger diffusion models. We conducted successful biasing experiments with three additional diffusion models: SD2.1, SDXL, and DeepFloyd IF.
* To strengthen the generality of our debiasing claims by auditing an additional debiasing method. We provided this (auditing Difflens), covering an additional domain of diffusion text-to-image debiasing methods that can be audited by our method.
* To test the applicability of our method in real-world settings by incorporating long-prompt and multiple-person generation. We added an experiment combining both, proving that our method can function in complex scenarios.
 * To ablate our method for additional LLM language priors, including a weaker, lower-parameter LLMs. We added experiments in biasing and auditing debiased models for two additional LLMs: Qwen3 8B and Llama 3.2 1B.
  * To add more race-biasing prompt and image generation examples of our method. This was also included.

Also for Reviewer jBhd, we address their requests:
* To showcase the generalization of our method to more modern, powerful diffusion models (as the aforementioned reviewer). We added biasing experiments for three more diffusion models SD2.1, SDXL, DeepFloyd IF, all within the initial scope of UNet-based architectures.
* To experiment with “less outdated models”. We henceforth tested both biasing and auditing performance of our method for two additional LLMs released in 2025 and included experiments with some of the latest and most widely used UNet-based diffusion models.
* To provide scientific insight into the nature of biases discovered by our method. We amended the main paper with a new section, where we study the generated prompts and deeply investigate how bias is introduced by our method.
* To update figures 1,2 for readability and ease of inspection. We further provide numerous additional qualitative examples of generated prompts and prompt-image pairs, which the reviewer stated are ”crucial for illustrating the method’s outputs”.

In addition to responding to these concrete requests, we have also added age-biasing experiments to address reviewer **Sk3D**’s comment on the generalizability of our method to novel attributes.
The remaining comments were by reviewer **Sk3D** on the naturalness of our prompts and by **Sk3D** and **jBhd** on the technical novelty of our method, which were addressed by extensive comments to the reviewers.

In conclusion, most of the requests involved concrete additions to the paper that were provided in the rebuttal; therefore, we are fairly confident that our responses addressed reviewers' concerns to a large extent. This indicates that our submission was, unfortunately, severely impacted by the discussion freeze.

---

### Meta-Review · Area_Chair_XarV · 2026-01-03

**Summary:**

The paper proposes Bias-Guided Prompt Search (BGPS), which uses an LLM together with attribute classifiers on diffusion activations to discover human-readable prompts that expose social biases in text-to-image models.

Reviewers agree the problem is important and that the method can indeed surface subtle and previously undocumented biases, often with more natural prompts than gradient-based alternatives. The authors significantly expanded experiments in the rebuttal (more diffusion models, debiasing methods, attributes, LLM priors, and more realistic prompt scenarios). However, the core technical ideas are largely incremental combinations of existing components, and the work offers limited new theoretical or conceptual insight into bias mechanisms or model behavior. Overall, I view the paper as a solid engineering effort but not sufficiently novel or deep to merit acceptance.

**Reviewer Concerns:**

The rebuttal addresses many concrete concerns: it broadens the experimental scope beyond SD1.5 to SD2.1, SDXL, and DeepFloyd IF, audits an additional debiasing method (DiffLens), adds age as a new attribute, evaluates multi-person and long prompts, and studies sensitivity to different LLM priors, including a smaller model. It also clarifies model versions, improves figure readability, and adds an analysis section on how bias is injected through lexical cues.

Nonetheless, key substantive concerns remain: the methodological novelty is modest (guided decoding with classifier feedback in a relatively standard form), there is no strong theoretical or analytical contribution, and many findings confirm well-known biases rather than delivering clearly new scientific insight. Questions about how “natural” the discovered prompts are in real user settings and about applicability beyond UNet-based architectures are only partially resolved. These remaining issues underpin my recommendation to reject.

**Reviewer Scores:**

Reviewer Sk3D: Likely unchanged (4), their main concerns about limited attributes, prompt naturalness, and modest technical novelty are only partially alleviated.

Reviewer zJ5g: Possibly unchanged (6), as their requests on scope and models were addressed, but broader concerns about generality and reliance on older architectures remain.

Reviewer jBhd: Likely unchanged (2). While some presentation and scope issues were improved, their core reservations about novelty and lack of new scientific insight are largely unaddressed.

---

### Decision · Program_Chairs · 2026-01-26

Reject